# Emission impacts of China's solid waste import ban and COVID-19 in the copper supply chain

John Ryter [1], Xinkai Fu[1], Karan Bhuwalka[2], Richard Roth[2] & Elsa A. Olivetti [1✉]

Climate change will increase the frequency and severity of supply chain disruptions and large-scale economic crises, also prompting environmentally protective local policies. Here we use econometric time series analysis, inventory-driven price formation, dynamic material flow analysis, and life cycle assessment to model each copper supply chain actor's response to China's solid waste import ban and the COVID-19 pandemic. We demonstrate that the economic changes associated with China's solid waste import ban increase primary refining within China, offsetting the environmental benefits of decreased copper scrap refining and generating a cumulative increase in $CO_2$-equivalent emissions of up to 13 Mt by 2040. Increasing China's refined copper imports reverses this trend, decreasing $CO_2e$ emissions in China (up to 180 Mt by 2040) and globally (up to 20 Mt). We test sensitivity to supply chain disruptions using GDP, mining, and refining shocks associated with the COVID-19 pandemic, showing the results translate onto disruption effects.

[1] Department of Materials Science and Engineering, Massachusetts Institute of Technology, Cambridge, MA, USA. [2] Materials Systems Laboratory, Materials Research Laboratory, Massachusetts Institute of Technology, Cambridge, MA, USA. ✉email: elsao@mit.edu

The transition toward a zero-carbon society is coupled with increasing electrification, prompting projections that demand for copper, the third most-consumed metal, will increase by >300% and consume ~2.5% of the world's energy by 2050, with greater increases under more equitable global development scenarios[1]. At the same time, copper ore grades continue to decline and extraction operations are increasingly concentrated in low-income regions with decreased enforcement of best practices[2,3]. These regions are also expected to experience the most intense effects of climate change[4], and copper resources in particular are concentrated in areas of high water scarcity risk[5]. These conflicting issues necessitate an integrated assessment of the copper material system and further emphasize the need for recycling and other resource efficiency principles in this supply chain[6].

In an effort to address air pollution and limit soil and water toxicity while maintaining economic progress, China has implemented resource efficiency policies centering the circular economy as a national development strategy[7,8]. While much of this legislation has garnered broad international support and improved local health outcomes[9,10], China's Green Fence (2013) and National Sword (2017) policies, which restrict nearly all solid waste imports, have also caused disruption across a variety of supply chains and led to increased landfilling and buildup of recyclables in high-income, waste-exporting countries[8,9,11,12]. Chinese companies facing consequent scrap supply shortages have reinvested in recycling facilities throughout Southeast Asia, Australia, and the United States[13–15], indicating a redistribution of scrap processing environmental impacts[16,17].

Studies to date have emphasized the ban's impact on plastic waste streams, primarily addressing geographical redistribution, increased landfilling, and environmental impacts[8,12,18,19]. Several authors have shown that China's domestic secondary copper supply is insufficient to meet its increasing metals demand, where Zeng et al., Wang et al., and Dong et al. explicitly account for changes in scrap imports[20–23]. With prior studies relying on top–down material flow analyses limited to China and primarily addressing potential scrap availability changes, a significant research gap remains in understanding global supply chain reactions stemming from the solid waste import ban, the resulting environmental impacts, and mechanisms for maximizing environmental benefits in China and globally. We demonstrate that the solid waste import ban causes increased primary refining and copper concentrate imports within China to account for refineries' loss of secondary material, generating effects throughout the material system that produce increasing environmental impacts both in China and globally.

The coronavirus disease 2019 (COVID-19) pandemic has introduced a supply chain shock alongside the solid waste import ban. Macroeconomic effects have decreased global copper demand[24–26], while recent outbreaks have increased death among miners and halted production at some of the world's largest copper mines, reduced refinery production, and limited scrap trade[27,28]. Simultaneous demand rebound in China is expected to produce supply deficits and price increases in 2021[27,29]. Relevant recent studies have addressed pandemic effects at the macroeconomic level, emphasizing short-term effects, and have been unable to comment on medium- to long-term impacts on individual supply chains, the impacts of combined supply–demand (SD) shocks, or external policy resilience to these shocks[30–32].

This study uses econometric time series analysis, inventory-driven price formation, dynamic material flow analysis (dMFA), and life cycle assessment (LCA) to estimate copper supply chain and environmental impact evolution under differing regional policy change and global economic shock scenarios. The model architecture (Fig. 1) differentiates supply chain behavior between China and the rest of the world (RoW) and considers differences in scrap composition, availability, and price in those two regions. We estimate the evolution of cathode and scrap prices, refinery processing fees (treatment and refining charges (TCRC)), and production and consumption of copper scrap, concentrate, and refined material through 2040, primarily reporting results as cumulative departure from baseline. The explicit modeling of mine-level opening, closing, and capacity utilization (CU) decisions; the economic modeling accounting for cascading effects throughout the copper supply chain; and the cost-driven optimization model determining scrap consumption changes between China and RoW highlight the contributions this methodology makes to material supply chain modeling.

## Results

**Current impacts of the China solid waste import ban.** China's major recent solid waste legislation includes the Green Fence action in 2013, the National Sword policy at the beginning of 2017, and the Implementation Plan on Banning Entry of Foreign Garbage and Reforming the Administrative System of Solid Waste Importation in July 2017. Recent policies affecting copper scrap include the announcement of the ban on Category 7 copper scrap in May 2017 and the imposition of tariffs on US copper scrap imports in August 2018. The ban on Category 7 copper scrap was implemented in December 2018 alongside the announcement of additional restrictions on Category 6 copper scrap imports, set to begin June 2019[23,33]. These policies have produced a redistribution of copper scrap trade and compositional changes in China's scrap imports (Fig. 2a). The gross weight of China's copper imports has declined over the past several years, with a corresponding increase in copper fraction producing a near-constant copper content by mass. These data reflect the success of China's policy goal of decreasing low-grade scrap imports and processing and coincide with the redistribution of global scrap trade and processing.

According to industry experts, Rep. of Korea, India, Germany, Taiwan, Belgium, Malaysia, Canada, and Vietnam have begun importing the majority of this newly available low-grade scrap, with some fraction simply being upgraded and re-exported to China (Supplementary Tables 9 and 10)[13,14]. This behavior is particularly evident in Rep. of Korea, Taiwan, Malaysia, Canada, and Indonesia; these regions have dramatically increased both copper scrap exports to China and copper scrap imports (Fig. 2b). Simultaneously, the fraction of US gross weight copper scrap exports going to China fell from 68 to 10% from 2017 to 2019, while the fraction of EU copper scrap exports to China fell from 29 to 15% in the same period. While the copper fraction of China's copper scrap imports nearly doubled from 2017 to 2019, within Indonesia, India, and Malaysia this value decreased 10, 15, and 39%, respectively, indicating more contaminated scrap streams. For the remaining nations listed above, the copper fraction of copper scrap imports changed no more than 5% over this period.

**Future impacts of the China solid waste import ban.** We simulate projected import restrictions for cases where scrap with less than 94 or 99% copper content would not be permitted for import. Such restrictions would eliminate imports of alloyed scrap grades, No.2 scrap, or both, with <99% copper content restricted and only No.1 copper scrap (Institute of Scrap Recycling Industries (ISRI) grade Barley) permitted to enter China. We analyze scrap import reductions of 50% year over year for each category with ±25% as sensitivity while other imports remain constant (Fig. 3a). Following China's reclassification of copper scrap to recycled copper or brass solid waste material on

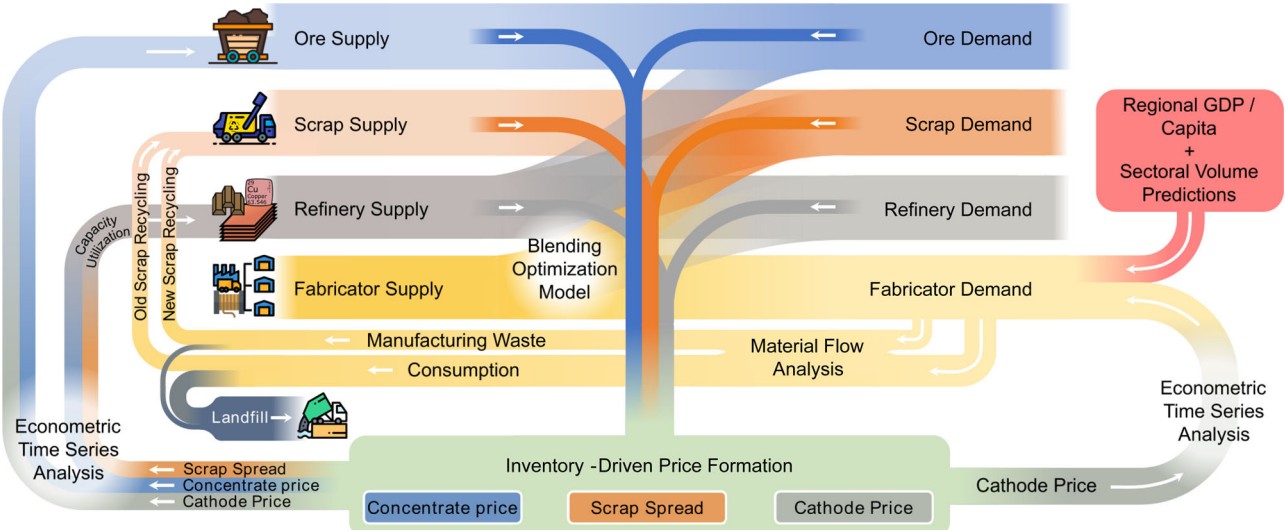

**Fig. 1 Model architecture.** Supply–demand imbalances for copper ore, scrap, and cathode are used to calculate treatment charges and refining charges (TCRC), scrap spread, and cathode price evolution, respectively, using inventory-driven price formation. Several of these supply–demand modules are linked; for example, refinery production (supply) uses both copper ore and scrap as raw materials and therefore determines demand for those materials. Likewise, fabricator production determines refined copper cathode and scrap demand through the blending optimization model. With prices formed, econometric time series analysis permits calculation of mine and refinery production evolution. Fabricator demand (consumption of copper) is determined using both cathode price and the exogenous variables gross domestic product (GDP) per capita and sectoral volume predictions. Fabricator demand permits calculation of scrap generation via material flow analysis. The resulting imbalances among these variables permit evolution year over year.

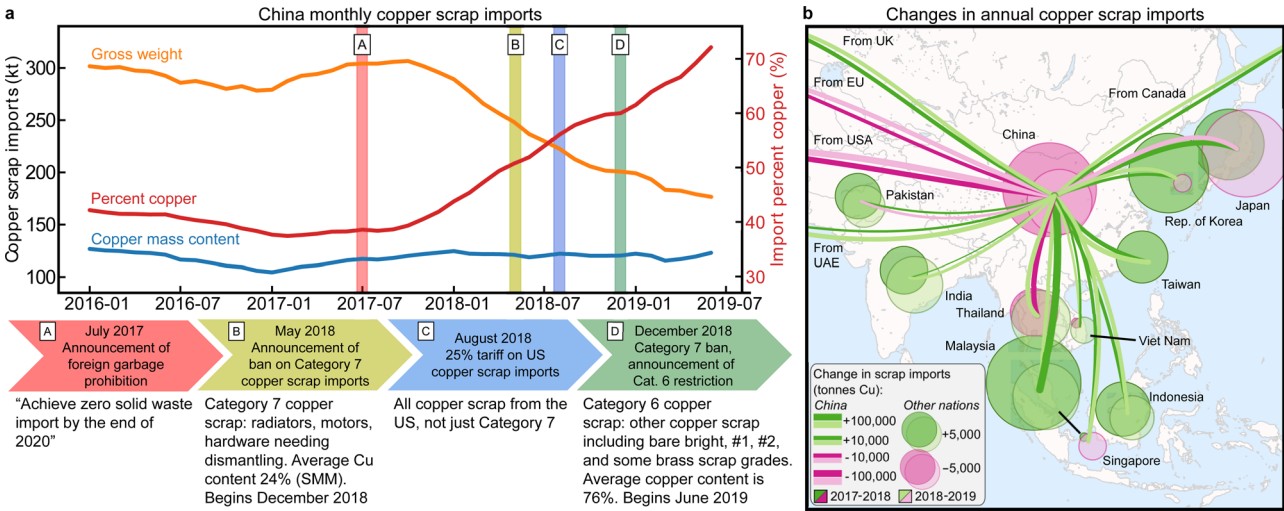

**Fig. 2 Changes in monthly and annual copper scrap imports for China and surrounding region. a** China's scrap-related policy actions and the consequent decrease in gross weight scrap imports (yellow), increase in the percent copper (red) for those imports, and the near-constant resulting copper mass content (blue). **b** Changes in the copper content of copper scrap imports for countries surrounding China and their exports to China, where year-over-year increases are shown in dark green (2017–2018) and light green (2018–2019), while decreases are shown in purple (2017–2018) and pink (2018–2019). Bubbles represent changes in the corresponding country's copper imports, with 2017–2018 always shown above and left of center and 2018–2019 below and right of center, while lines represent changes in that country's exports to China. Bubble diameters and line widths vary on logarithmic scales, with values reported in Supplementary Table 9. This figure is created using the matplotlib Basemap package in Python[75,76]. Underlying data used to create this figure may be found in a data repository at https://doi.org/10.6084/m9.figshare.14390489.v3.

July 1, 2020, the No.2 ban scenario approximates reality; industry experts predict free trade of high-quality recycled copper and brass raw materials and consequent policy stability in the near future[34]. We provide additional granularity surrounding the No.2 scrap ban scenario.

In an effort to highlight relative changes, results are presented as the cumulative departure from the baseline scenario as a percentage of the projected 2020 value for that parameter;

absolute responses (in thousand metric tonnes) may be found in Supplementary Figs. 5–7. The solid waste import ban shifts scrap availability from China to RoW, increasing prices in China and decreasing prices for RoW, prompting a redistribution of primary and secondary refining production (Fig. 3b). Due to time delays and differing magnitudes of scrap availability shifts, the decrease in RoW primary refining and increase in RoW secondary refining lag the opposing responses in China, producing a global increase

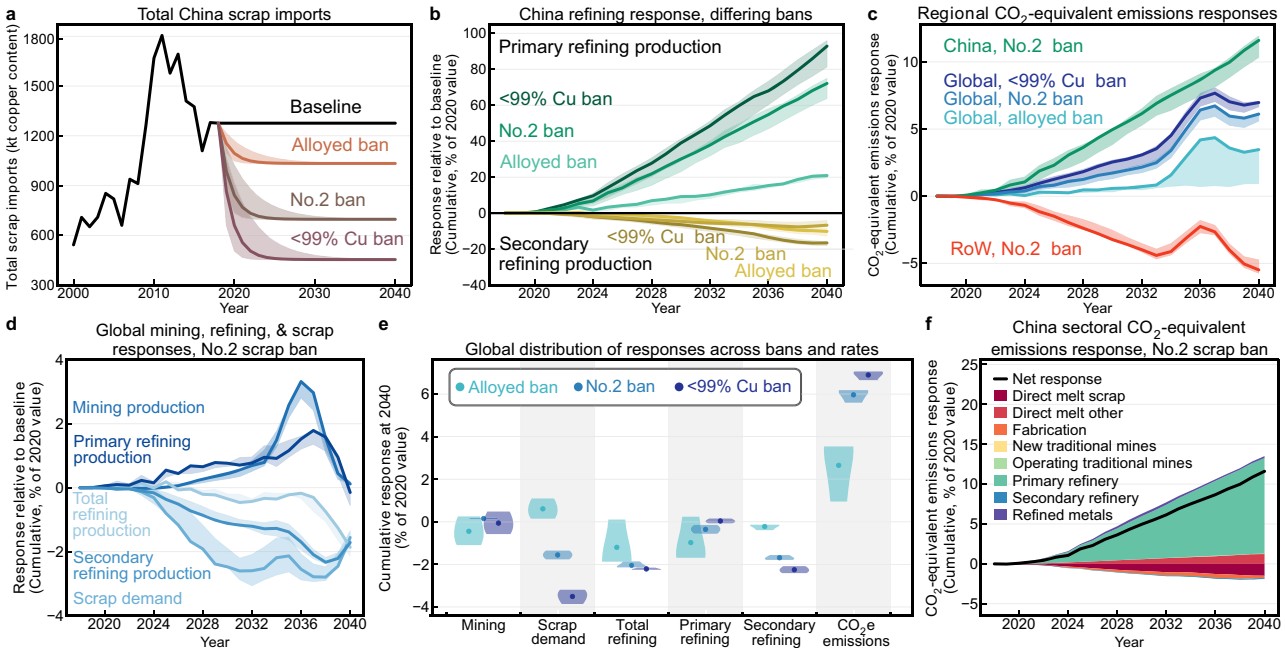

**Fig. 3 System response for China's solid waste import ban, No.2 ban with additional emphasis.** In each subfigure, the shading surrounding solid lines corresponds to the variation in ban rate: 25, 50, or 75% reduction in China's scrap imports year over year, where solid lines represent a 50% year-over-year ban of the listed scrap type. **a** Scrap imports for each scenario in copper content. **b** Cumulative primary and secondary refining responses in China relative to baseline for each scenario using total refining production as reference. **c** Cumulative global $CO_2e$ emission responses for each scenario, with China and RoW responses for the No.2 scrap ban only. Global values are used as divisors for regional parameters for ease of comparison. **d** Cumulative global mining, refining, and scrap demand responses for the No.2 scrap ban over the simulation period. Labels correspond with relative positions of their values at 2040. **e** The distributions of global mining, scrap demand, total refining, primary refining, secondary refining, and $CO_2e$ emissions responses for each ban, evaluated cumulatively at 2040 relative to baseline. Points represent the mean of the three ban rates, while the shaded regions represent the distributions of ban rate results. **f** Cumulative sectoral $CO_2e$ emission response for China, where all increasing impacts were plotted above the x-axis, all decreasing impacts were plotted below the x-axis, and the black line represents the net response within China as a result of these sectoral changes. Underlying data used to create this figure may be found in a data repository at https://doi.org/10.6084/m9.figshare.14390489.v3.

in primary refining production and decrease in secondary refining production (Fig. 3d). Increasing scrap demand in RoW is unable to offset the increasing prices and decreasing demand in China, generating a net decrease in global scrap demand. Resulting changes in cathode price are insufficient to drive a significant departure from baseline and the small increases in mining and primary refining production are erased by market corrections by 2040. While scrap consumption appears approximately linear as a function of scrap quantity restricted, the No.2 ban generates a larger system response than the ban on alloyed scraps for refining, mining, and $CO_2e$ emission responses (Fig. 3e).

Global parameters either decrease or remain effectively unchanged relative to baseline by 2040, leading to the expectation of negligible changes in environmental impacts. However, the redistribution of primary and secondary refining between China and RoW, coupled with higher unit impacts of China's primary refining, produces an increase in global $CO_2e$ emissions of 13 Mt, or 6.1% of current annual emissions due to copper production (Fig. 3c). This result corresponds with a 29% (25 Mt, 12% of global) increase in $CO_2e$ emissions from copper within China, equivalent to the annual $CO_2$ emissions of 5.4 million gasoline vehicles. Without the significant decarbonization of China's electricity grid predicted by the U.S. Energy Information Administration's reference case[35], global emissions increase by as much as 35 Mt by 2040. This large emissions increase is the result of increased primary refining production, while the decreases in scrap use and fabrication account for the smaller contributions toward decreasing emissions (Fig. 3f). Increasing

primary refining production requires increased concentrate imports—determined endogenously—due to limited domestic copper ore bodies, continuing the present logistic growth trend (Supplementary Fig. 3).

All 12 environmental impact indicators considered in this study follow trends similar to the $CO_2e$ emission response (Supplementary Figs. 8 and 9). Given the pollution reduction goal of China's solid waste policies, smog-, respiratory-, and human toxicity-related emissions are of particular importance and these results show 34% (5.5 Mt $O_3$ eq), 53% (450 kt PM2.5 eq), and 44% (1.2 million comparative human toxicity units (MCTUh)) increases above the baseline 2020 value within China. Without further action, economic responses to this policy will produce unintended negative environmental consequences.

**Responding to the China solid waste import ban**. Higher Chinese refined copper imports may mitigate the effects of the scrap ban and take advantage of the newly available scrap available outside China and correspond with some of China's foreign investment strategies to date[8,14,15,36]. We increase or decrease China's refined copper imports at rates of 100 or 200 kt/year (Fig. 4a) coincident with the No.2 scrap ban. We assumed that China would not begin exporting refined copper and the minimum value was set to zero. We show that increasing China's refined copper imports redistributes both primary and secondary refining production from China to RoW by shifting regional refined copper demand (Fig. 4c). Globally, this shift enables better use of displaced scrap material (cumulative 10% of 2020 value,

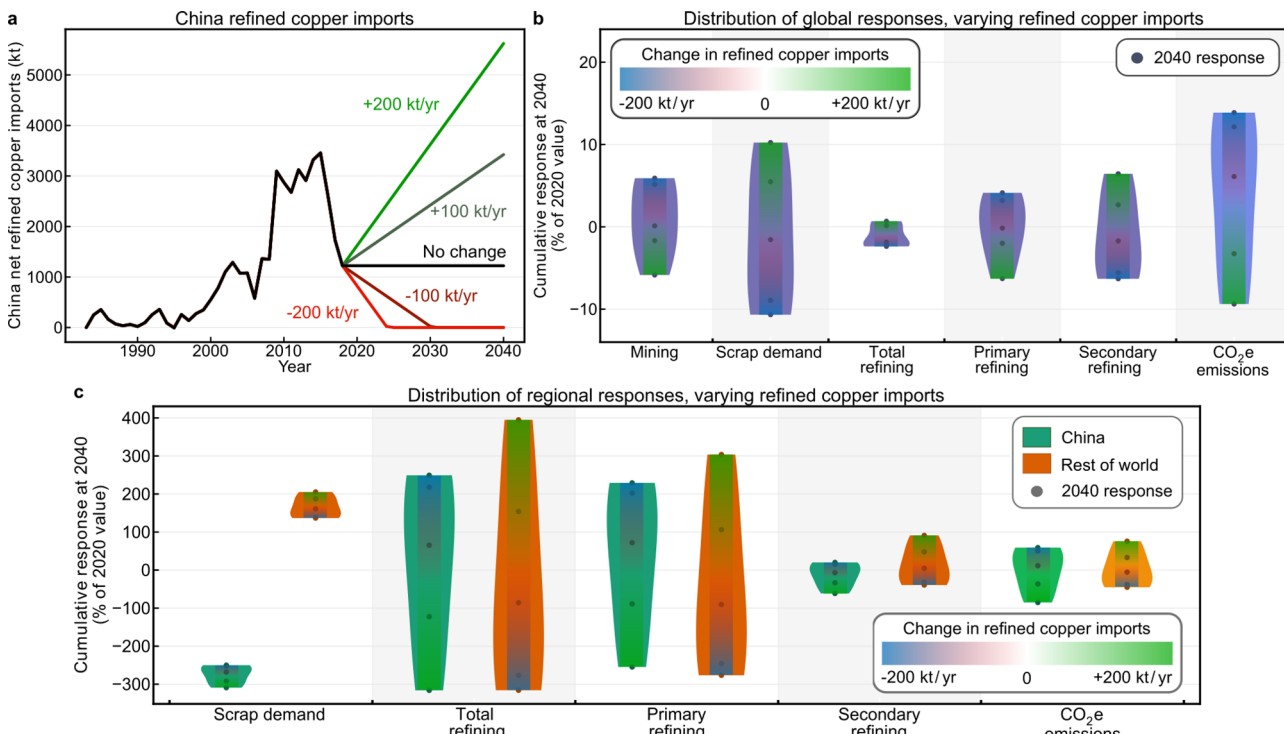

**Fig. 4 Results of changing China's refined copper imports alongside the No.2 scrap ban. a** Historical Chinese net refined copper imports and scenario definition, where refined copper imports are increased or decreased at rates of 100 or 200 kt/year, with minimum value zero. **b** Violin plot showing the distributions of global responses for varying refined copper imports, where $CO_2e$ emissions are highlighted as an aggregate response. Points show the values for each of the five scenarios and color bar indicates directionality. Primary, secondary, and total refining use the total refining value as divisor. **c** Violin plot showing the distributions of regional responses for each supply chain actor, with $CO_2e$ emissions highlighted as a system-level response. Points show the values for each of the five scenarios and color bar indicates directionality. China is shown in green, RoW in orange. $CO_2e$ emissions use the global value as divisor for ease of comparison; the extrema using local divisors are China: −270% and 170%, RoW: −120% and 210%. Primary, secondary, and total refining use the regional total refining value as divisor. Underlying data used to create this figure may be found in a data repository at https://doi.org/10.6084/m9.figshare.14390489.v3.

1.0 Mt), resulting in increased secondary refining (6.4%, 1.3 Mt), decreased primary refining (6.3%, 1.3 Mt), and decreased mining production (5.9%, 1.2 Mt; Fig. 4b).

This global decrease in primary refining and mining production (Fig. 4b), coupled with the redistribution of refining from China to RoW (Fig. 4c), enables a cumulative global reduction in $CO_2e$ emissions equal to 9% of 2020 copper production emissions. This value exceeds that of the mining reduction alone due to the differences in unit environmental impacts of refining within China and the global average. Within China, $CO_2e$ emissions decrease a cumulative 210% (180 Mt or 87% of global) of the estimated 2020 emissions in China by 2040, smog-related emissions decrease 215% (34 Mt $O_3$ eq.), respiratory-related emissions decrease 250% (2.1 Mt PM2.5 eq.), and human toxicity-related emissions decrease 220% (6 MCTUh). These values assume significant decarbonization of global electrical grids in the baseline scenario, particularly in China. Given their dependence on the relative unit impacts of primary and secondary refining in and outside of China, emission reductions relative to baseline are greater still if China's carbon intensity of electricity remains high or RoW implements similarly ambitious decarbonization.

Decreasing China's refined copper imports represents a continuation of the 2015–2018 trend, while increasing imports represents the case where China acts to redistribute primary refining activities outside its borders alongside scrap refining activities. Further decreasing China's refined copper imports exacerbates the negative effects of the scrap ban by causing China to increase refining further still, while reversing that trend produces environmental benefits both within China and globally.

Increasing China's refined copper imports does, however, increase environmental impacts for RoW. While much of the redistribution to date has been to regions with reduced environmental regulations, these localized impacts can be minimized if smelting and refining investment prioritizes regions and technologies with better environmental practices and electrical grid emissions intensities. The relative inelasticity of total refining production, the nearly equal and opposite changes in global mining production, and scrap consumption stemming from increasing China's refined copper imports indicate a market-stable transition in the direction of a circular economy and at minimum lower unit emissions for the copper material system (Fig. 4b).

**Sensitivity to supply chain disruptions.** To understand how these policies evolve in the face of future climate- and social unrest-induced supply chain disruptions, we simulate the impact of major system shocks using production and consumption changes stemming from the COVID-19 pandemic. We use 2019–2021 gross domestic product (GDP) changes from the International Monetary Fund (IMF; Fig. 5a) and calculate copper demand endogenously from GDP per capita evolution (Fig. 5b). We account for operational discontinuities by adjusting mine CU, refinery CU, and refinery secondary ratio (SR; the fraction of secondary material used in secondary refineries) for 2020 according to data from the International Copper Study Group (ICSG). We use mean year over year changes from the first 5 months of 2019 and 2020 ±50%, producing large, moderate,

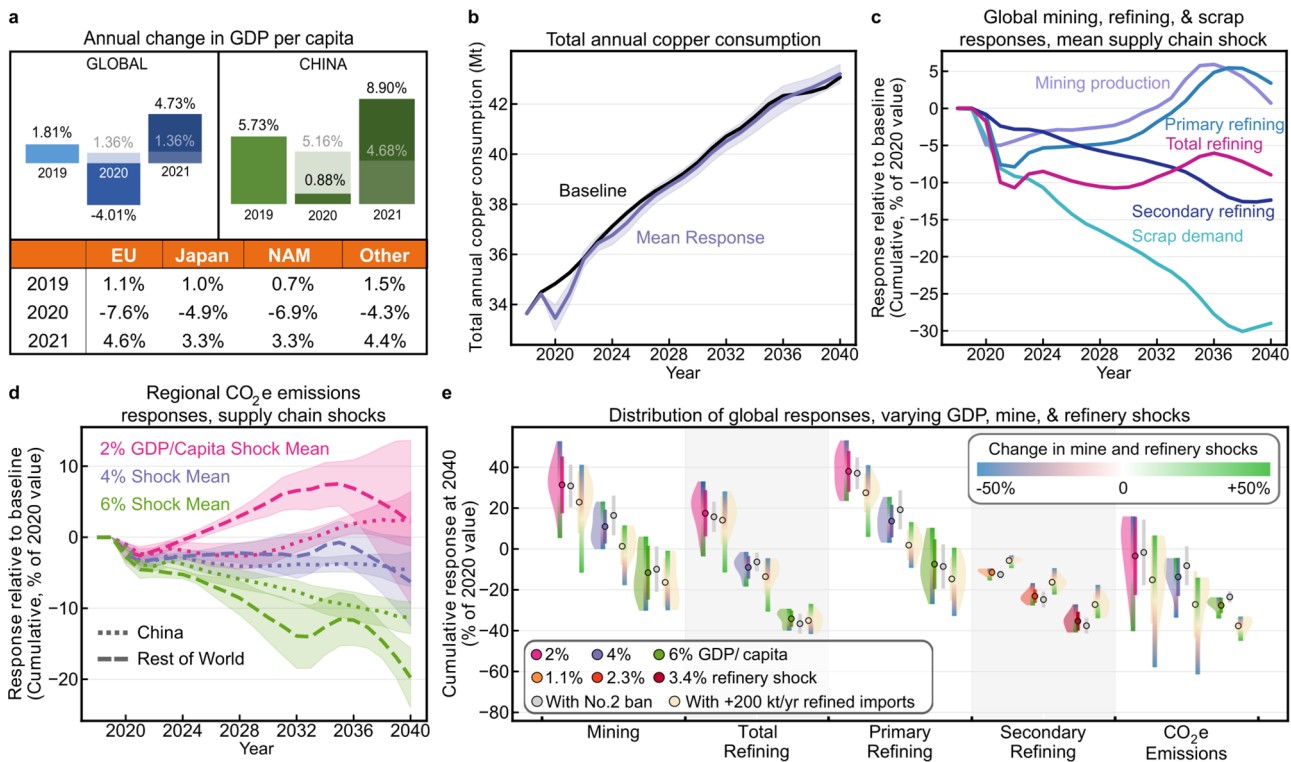

**Fig. 5 System response to COVID-19 scenarios and supply chain disruption sensitivities. a** The annual changes in gross domestic product (GDP) per capita for 2019, 2020, and 2021 used in the COVID-19 response scenario, using values adapted from annual change in GDP from the International Monetary Fund. Baseline values for China and global are shown in gray, while the table shows projections for the European Union (EU), Japan, North America (NAM), and other. **b** Global annual copper consumption including alloyed and unalloyed refined and scrap copper consumption for baseline and the mean COVID-19 scenario response. Shaded areas represent one standard deviation from the mean. **c** Cumulative global secondary refining, scrap demand, mining production, total refining, and primary refining responses relative to baseline as a percent of the 2020 value, labeled from top to bottom using 2040 as reference. Standard deviations are not shown for clarity, but are included in the data repository described below. **d** Cumulative $CO_2$e emission responses relative to baseline for China and RoW as a percentage of the 2020 global value. Shaded areas represent one standard deviation from the mean. **e** Violin plot showing the distribution of global responses for COVID-19 response scenarios using 2, 4, and 6% declines in global GDP per capita from 2019 to 2020, with mean values shown as same colored points. Gray points and bars represent the mean system response and standard deviation when the No.2 scrap China solid waste import ban is simulated simultaneously with the COVID-19 shocks, and light yellow incorporates the 200 kt/year increase in China's refined copper imports alongside the No.2 ban. Green and blue color bar overlays indicate the magnitude of the mine and refinery system shocks. For secondary refining, GDP changes produced near-equal violin plots and here the data are grouped by the three levels of refinery shock instead, with the color bars representing changes in mine and GDP/capita shocks. Underlying data for this figure may be found in a data repository https://doi.org/10.6084/m9.figshare.14390489.v3.

and small responses for each of these four parameters. The annual timescale of this model necessitates that evolution of copper price and other market indicators begins in 2021, the year following the shock. Nonetheless, the near-term evolution of the demand, primary and secondary refining production, mine production, and cathode price align well with projections from Roskill[37], ICSG[38], S&P Global[29], and GlobalData[39]. See Supplementary Methods: Near-term system response to COVID-19 shocks.

In testing supply chain resilience to such shocks, we found a reduction in cumulative total refining production (Fig. 5c), indicating that post-shock recovery effects may not sufficiently compensate for short-term production disruption. Demand rebound prompts a period of high copper cathode prices, incentivizing mining and primary refining production at the expense of scrap consumption and secondary refining production (Fig. 5c). China exhibits a more muted $CO_2$e emission response due to its more limited GDP growth reductions and low fraction of global mining production, with the bulk of China and RoW declines attributable to manufacturing contraction (Fig. 5d). Among small GDP shocks, incorporating larger mine and refinery shocks generates $CO_2$e emission reductions equivalent to those of large GDP shocks (Fig. 5e). Further analysis has shown

that these reductions stem from the SR shock specifically (Supplementary Figs. 14 and 15). This shock produces an increase in primary refining production, leading to a copper concentrate supply deficit, mine overproduction, and eventually a price collapse that promotes mine closure and delays mine opening, while limiting secondary refining recovery. Changes in GDP per capita dominate the long-term system response, with mine and refinery production disruptions producing small but non-negligible changes and SR changes dominating for sufficiently small GDP shocks.

Incorporating these supply chain shocks immediately following the No.2 scrap ban generates combined effects approximating the sum of the independent responses. Relative to the no-ban scenario described above, secondary refining production decreases while primary refining production increases or remains constant due to time lag effects associated with China–RoW refining redistribution. These changes lead to increases in cumulative mean $CO_2$e emissions of 3.7–12 Mt (1.7–5.5% of the 2020 value) relative to the supply chain shock-centered responses, with mean values taken for each GDP/capita shock and including all refinery and mine shocks. These values approach the 13 Mt value found without supply chain shocks. Within China, mean

emissions increase 22–36 Mt (10–12% of global 2020 value, 24–29% of China 2020 value) relative to the supply chain shock-centered scenarios, overlapping the 25 Mt increase in the no-shock scenarios above. Relative to baseline, mean emissions in China increase by up to 0.8–26 Mt. These results indicate that, within China, the environmental benefits stemming from these supply chain disruptions are insufficient to offset the increasing emissions resulting from the solid waste import ban even for the largest economic and trade disruptions.

When refined copper import changes are coincident with the solid waste import ban and supply chain disruptions, we again observe a redistribution of primary and secondary refining. These changes result in decreased primary refining and mining production, increased secondary refining and scrap consumption, and near-constant total refining production indicating a decrease in copper system unit impacts amid business-as-usual demand growth following the shock. For a large increase in China's refined copper imports (+200 kt/year), mean global $CO_2e$ emissions decrease a cumulative 22–29 Mt (10–13% of 2020 value) relative to supply chain shocks alone (compared with 20 Mt decrease without supply chain shocks above), or 33–81 Mt relative to baseline, doubling even the largest shock-induced emission reductions. These impact reductions are concentrated within China, with China's cumulative $CO_2e$ emissions decreasing ~190 Mt (90% of global 2020 value, 210% of China 2020 value) relative to supply chain shocks alone or 180–210 Mt relative to baseline (39–46 million vehicles).

For both the scrap and refined copper import policies explored here, their modeled impacts manifest as an approximately additive effect when combined with the supply chain disruption scenarios. The modeled responses therefore remain valid for large-scale supply chain shocks, including those that vary in regional severity, given that the GDP per capita shock is more intense in RoW than in China. Large supply chain shocks such as those associated with the COVID-19 pandemic may partially mask the environmental impacts associated with the solid waste import ban, but it will not erase them, and a redistribution of primary refining activity to RoW remains a viable mechanism for mitigating these effects. Under circumstances where these shocks produce additional restrictions on environmentally harmful industries[26], environmental impact reductions beyond those shown here may occur.

## Discussion

China's solid waste import ban has induced a shift in the location of scrap processing, with Malaysia and other Southeast Asian nations accounting for the majority of recent scrap processing investment (Fig. 2b). With the bulk of scrap being upgraded and re-exported to China, it is clear that these nations are not bene-fiting from the efficient allocation of raw materials typically ascribed to the free trade of solid waste[40]. While economy-scale analyses provide evidence for the environmental Kuznets curve—that per-capita emissions follow an inverted U-shaped trajectory as per-capita income increases—accounting for trade has been shown to produce a linearly increasing curve instead[41]. These results, in combination with the estimated future impacts of the solid waste import ban, provide the first system-level evidence supporting these conclusions. When increasing China's refined copper imports, ~80% of the emissions reduction within China is redistributed to RoW, generating a net global decrease in emissions due to economic effects and lower unit impacts in RoW. This value is consequently dependent on the relative environmental impacts of industrial activities for each trade partner, explicitly demonstrating that individual nations' environmental impact reductions may be directly accomplished via burden

shifting. To maximize the benefits of this burden shifting, new refineries must be constructed with best available technologies in place and located within low-emission electricity grids.

Zeng et al.[20], Dong et al.[23], Eheliyagoda[42], Liu et al.[17], and Wang et al.[21] show that China's domestic copper scrap generation cannot meet its increasing raw material demand and we confirm these results. We also demonstrate that the solid waste import ban results in increased primary refining and concentrate imports within China to account for refineries' loss of secondary material, generating effects throughout the material system that produce increasing environmental impacts. The solid waste import ban's impacts on scrap availability, and consequently regional prices, drive a redistribution of primary refining from RoW to China and of secondary refining from China to RoW. Even as China's scrap- and manufacturing-related emissions decrease, increasing primary refining production offsets these benefits (Fig. 3f) and generates a net increase in environmental impacts by 2040 across all impact categories considered in this study, both in China and globally (Fig. 3e). The RoW response rate keeps the redistribution of refining activities from being even and proportional, and secondary refining is projected to decrease globally. The RoW must act quickly to limit this redistribution and develop low-impact secondary processing and refining capacity such that the environmental impacts of the copper supply chain do not increase further still.

Increasing China's refined copper imports acts to mitigate these effects and capitalizes on the newly available scrap outside China (Fig. 4b), mirroring some of China's current foreign investment strategies[8,14,15,36]. The nearly equal and opposite resulting changes in global mining production and scrap consumption indicate a market-stable transition toward a circular economy and lower unit emissions for the copper material system. Given their compositional similarities and that this study includes smelting within refining processes, model results would be similar for increased imports of copper blister, anode, or fabricated products. Potential mechanisms for increasing imports of these materials therefore include limiting China's concentrate imports below 10 Mt (see Supplementary Fig. 3) and increasing China's refined copper imports above 2018 levels. To limit the adverse environmental impacts of the solid waste import ban, China could prioritize investment in increasing domestic scrap collection and the implementation of refining and fabrication technologies with best available techniques.

Our analysis of global economic and supply chain shocks shows that the long-term environmental impacts of the solid waste import ban and refined copper import policies remain valid even with disruptions in mining and refining production. Elevated sensitivity exists for changes in GDP growth and refinery SR, where GDP growth shocks dominate at high values. We also show that the impacts of policies such as the China solid waste import ban and increasing China's refined copper imports translate approximately linearly onto such shocks, indicating the results of these policies are robust in the face of supply chain disruption. Additionally, we observe that, because mines respond to longer-term market trends, primary production is more robust to these supply chain shocks than secondary production. The resulting decline in secondary demand indicates that further emission reductions could be enabled by implementing policies supporting the recovery of secondary markets.

Future research surrounding regional differences in refinery and scrap price behavior, scrap import–export compositions, future copper mining production, and mine opening and closing behavior would address model assumptions and reduce the associated uncertainty. Additional geographical granularity would permit a coincident increase in the degree of localization accessible to the model, both in terms of policies and associated

**Table 1 Scenario descriptions.**

| | Scenario | Description |
|---|---|---|
| Future impacts of the China solid waste import ban | Alloyed scrap ban | Alloyed scraps are barred from import to China, leading to 25, 50, or 75% declines in alloyed scrap imports relative to the previous year, starting in 2019. No.1 and No.2 copper scrap imports remain constant. Decline of 50% year over year was selected as the mean ban rate, with 25 and 75% included as sensitivities |
| | No.2 scrap ban | Scraps requiring refining are barred from import to China, leading to 25, 50, or 75% declines in No.2 scrap imports relative to the previous year, starting in 2019. No.1 and alloyed copper scrap imports remain constant. Decline of 50% year over year was selected as the mean ban rate, with 25 and 75% included as sensitivities |
| | <99% Cu scrap ban | Alloyed scraps and scrap requiring refining are barred from import to China, leading to 25, 50, or 75% declines relative to the previous year, for all scrap imports except No.1 copper scrap, starting in 2019. No.1 scrap imports remain constant. Decline of 50% year over year was selected as the mean ban rate, with 25 and 75% included as sensitivities |
| Responding to the China solid waste import ban | Change in China's refined copper imports | China's refined copper imports change by −200, −100, 0, 100, 200 kt/year relative to the prior year's value, starting in 2019. We assume China does not become a net exporter of refined copper and negative refined import values are not permitted. In scenarios without refined copper import changes, China's refined copper imports remain constant at the 2018 level |
| Sensitivity to supply chain disruptions | GDP/capita reduction | Due to the economic disruption associated with COVID-19, changes in GDP/capita relative to the prior year were evaluated in line with Fig. 5a, where the global reduction from 2019 to 2020 was 2, 4, 6%, increasing 2.35, 4.7, 7.05% from 2020 to 2021. Mean changes were −4 and 4.7% for 2019–2020 and 2020–2021, respectively. All years beyond 2021 use baseline GDP/capita evolution |
| | Refinery shock | Due to factory closures and shipping restrictions associated with COVID-19, two simultaneous changes in refinery operation occurred, respectively. First, refinery capacity utilization was reduced by 1.14, 2.29, 3.93% for 2019–2020, with mean value 2.29%. Second, shipping restrictions reduced scrap use, causing refinery secondary ratios to decrease 3.46, 6.93, 10.4% for 2019–2020, with 6.93% used as mean. Both these shocks are implemented simultaneously except in Supplementary Fig. 15, where the secondary ratio shock was removed to highlight result dependence on this component of the refinery shock over the capacity utilization reduction. For 2021 onward, and the rest of the simulation, these parameters were not constrained and evolve in accordance with the model |
| | Mine supply shock | Due to the suspension of mining operations due to COVID-19 outbreaks, mining capacity utilization decreased 1.31, 2.62, 3.93% globally, with mean 2.62%. For 2021 onward, mine capacity utilization changes were not constrained and evolve in accordance with the model |

Unalloyed scrap grades include No.1 and No.2 copper scrap, while alloyed scrap grades include yellow brass, leaded yellow brass, red brass, leaded yellow brass, cartridge, manganese bronze, nickel silver, ocean, aluminum bronze, tin bronze, and leaded tin bronze.

environmental impacts. Alternative environmental impact reduction strategies may be explored as scenarios, including carbon pricing, minimum recycled content policies, and material lifetime extension. The regional aspect of this model enables comparison of local and global variations on such policy, further informing the benefit of global efforts toward emission reduction.

## Methods

**Trade data**. Import data for each country were obtained from UN Comtrade using HS commodity code 7404, "copper; waste and scrap."[43] We subtracted exports from imports to calculate net imports for each region and its trade partners, using individual trade partners for China and the global sum for all other countries. Copper content values were calculated using the reported monetary value of the trade and year average London Metal Exchange (LME) copper cathode price. We considered changes in net import copper content from 2017 to 2018 and from 2018 to 2019, selecting countries with 2017–2018 changes of >300 tonnes, excluding those with unavailable 2018–2019 data with the exception of Taiwan. Monthly China copper scrap import data for Fig. 2a were obtained from Big Trade Data (www.bigtradedata.com), with copper content calculated using the monetary value of trade and the monthly LME copper cathode price.

**Scenarios and scenario implementation**. To evaluate the future impacts of the China solid waste import ban, methods for response, and sensitivities to supply chain disruptions such as COVID-19, several exogenous and endogenous model parameters were constrained. In the absence of scrap and refined import policy changes, China's scrap and refined copper imports were held constant at the 2018 values, with scrap imports broken into 14 categories representing the most common ISRI post-consumer scrap grades as described below. The scrap import distribution among these categories was assumed equivalent to that of old scrap generation in RoW in that year. In the absence of COVID-19-related shocks, GDP/capita changes proceed according to IMF projections made prior to any COVID-19 outbreaks, while refinery and mining operations evolve according to their relevant elasticities. Scenario definitions are given in Table 1.

The range of scenarios explored here reflects the uncertainty surrounding the potential market impacts of COVID-19, including cases of supply surplus and deficit and the resulting reverberations through the material system. These changes included GDP/capita changes in 2020 and 2021, mining CU change in 2020, and refinery CU and SR changes in 2020. Each change was implemented in the model by manually inserting the respective value, rather than using its exogenous (GDP/capita) or endogenously determined (all others) value. GDP/capita reductions result from the economic shock induced by COVID-19, while the remaining changes stem from supply chain interruptions associated with viral outbreaks[28]. Outside these years, the SD imbalances emanating from these shocks were permitted to evolve in accordance with the model.

GDP/capita changes were implemented in agreement with Fig. 5a, with mean global reduction in GDP/capita of 4% from 2019 to 2020 followed by a 4.7% increase from 2020 to 2021[44]. Refinery operations were affected both by plant closures and slowdowns in scrap collection, processing, and transportation, resulting in CU falling 2.29% for 2019–2020 and SR falling 6.93%[28]. Mining operations were suspended in efforts to slow or avoid outbreaks, resulting in mine CU falling 2.62% for 2019–2020[28]. Sensitivities were explored by adjusting the values above ±50%. The resulting near-term changes in demand, primary and secondary refining production, mine production, and cathode price are detailed in Supplementary Methods: Near-term system response to COVID-19 shocks and align with projections from Roskill[37], ICSG[38], S&P Global[29], and GlobalData[39].

**Model framework**. The copper material system model described in previous work formed the basis of this model. The production and consumption of four material stages within the copper system—ore, scrap, refined copper cathode, and semi-fabricated goods—were modeled based on MFA and inventory-driven price formation, where econometric time series analysis of historical data was used to determine the price, production, and consumption responses that minimize SD imbalances for each material stage. As such, the original model flow can be characterized as follows, starting from model initialization in 2018 with iteration on an annual basis through 2040:

1. Cathode price, TCRC, and scrap spreads (differences between scrap and cathode prices) are determined based on SD imbalances from the previous year;
2. Traditional (concentrate) and solvent extraction and electrowinning (SX-EW) mines respond to the market state—cathode price and TCRC—by altering their CU, opening, or closing;
3. Total copper demand is estimated based on exogenous GDP per capita and sectoral (e.g., construction, automotive, or industrial) volume projections, coupled with copper intensity (kg Cu per kg product) response to price, as developed from collaborator Karan Bhuwalka[45];
4. Primary and secondary refinery production are estimated based on TCRC and No.2 (ISRI code Birch) scrap spread. Smelters are included within the refinery module;
5. Post-consumer (old) and post-industrial (new) scrap supply are estimated using standard dMFA procedure, using previous years' demand values, lognormally distributed (lognormal($\mu$, $\sigma^2$) with $\mu = 0.1\sigma$) sectoral product lifetimes, estimated scrap collection rates, technical recycling efficiencies, and home and exchange scrap ratios from Glöser et al.[46], SMM[33], and previous work[47];
6. Refined copper and direct melt scrap consumption are estimated based on the prior demand prediction, the ratio between alloyed and unalloyed copper products, and scrap conversion efficiencies.

Several components required further development to enable regional and scrap composition considerations for system evolution under the China solid waste import ban. The prior global model was separated into two co-evolving components—China and the RoW—where the distribution of refined and scrap material consumption between these regions was determined by a linear programming optimization model. For this optimization model to operate, additional granularity surrounding semi-fabricator compositional requirements and scrap composition, availability, and demand were required. Each of these model expansions are described below. A brief discussion of the model base is also given, with additional detail in previous work. A gate-to-gate LCA was performed using the production data resulting from scenario analysis. A comprehensive outline of model iteration is shown in Supplementary Fig. 16.

**Price formation**. In all cases, prices are adjusted for inflation. Cathode price evolution is based on the balance between cathode supply (refinery production and SX-EW mining production) and demand (semi-fabricator cathode consumption). We used spot price rather than future contract data because spot price is believed to be a better indicator of SD balance than future price for commodities[48–50]. With minimal difference between commodity exchange prices due to arbitrage[51], we used only LME cathode spot price in developing this model due to its higher historical liquidity. Factors influencing copper cathode price have been studied extensively[52–55], with short-term changes particularly vulnerable to factors outside SD balance. We focus only on the effects of SD imbalance, developing an autoregressive distributed lag model that uses oil price to account for other market effects, with more details in prior work[47,56]. Cathode price then evolves according to Eq. (1).

$$P_{\text{cathode},t} = P_{\text{cathode},t-1} - 0.461(S_{\text{cathode},t-1} - D_{\text{cathode},t-1}) \quad (1)$$

where $P_{\text{cathode},t-1}$, $S_{\text{cathode},t-1}$, and $D_{\text{cathode},t-1}$ represent cathode price, supply, and demand in the preceding year, respectively. TCRC are a component of mining cash costs and a source of revenue for smelters and refineries, serving as an indicator of copper concentrate SD imbalance. With annual TCRC determined by negotiation among the world's largest smelting and mining corporations and spot TCRC highly correlated with annual TCRC, we assumed that annual TCRC is more representative of the concentrate market[56]. An ordinary least squares (OLS) regression model was constructed using SNL and ICSG data from 1982 to 2018[57,58], resulting

in TCRC evolution following Eq. (2).

$$\text{TCRC}_t = \text{TCRC}_{t-1} + 0.164(S_{\text{Conc},t-1} - D_{\text{Conc},t-1}) \quad (2)$$

where $\text{TCRC}_t$ is the annual TCRC of year $t$ (in USD/t payable copper), $S_{\text{Conc},t-1}$ is the world concentrate supply (concentrate mine production) in year $t-1$, and $D_{\text{Conc},t-1}$ is concentrate demand (primary refinery concentrate consumption) in year $t-1$.

Due to the high correlation between cathode price and scrap price, as well as the expectation that scrap SD balance impacts only the price of scrap relative to cathode, we model the difference between cathode price and scrap price, the scrap spread. With SD data lacking for individual scrap grades, we estimated spread evolution based on the imbalance of total scrap supply and demand. With No.2 copper scrap (ISRI grade birch)[59] the most significant predictor within other modules, we present only the No.2 scrap results here. We developed an OLS model using monthly scrap price data (1995–2018) from Fastmarkets AMM, resulting in No.2 spread evolution according to Eq. (3). We assumed that the SD effect was equal to the cathode SD effect on No.2 spread via cathode price effects (cathode SD effect on cathode price multiplied by the cathode price effect on No.2 spread)[60]. This assumption was explored using sensitivity analysis in previous work[47,56].

$$\text{Spread}_{\text{Birch},t} = \text{Spread}_{\text{Birch},t-1} + 0.184\Delta P_{\text{Cat},t} + 0.0845(S_{\text{Scrap},t-1} - C_{\text{Scrap},t-1}) \quad (3)$$

where $\text{Spread}_{\text{Birch},t}$ is the No. 2 spread at time $t$, $\Delta P_{\text{Cat},t}$ is the change in cathode price from year $t-1$ to year $t$, $S_{\text{Scrap},t-1}$ is the scrap supply in year $t-1$, and $C_{\text{Scrap},t-1}$ is scrap consumption (demand) in year $t-1$.

**Primary supply module**. We model evolution of individual mines, with currently operating mines from SNL and an incentive pool of potential mines created by resampling with perturbation mines that have opened during 2015–2018. Ore grade elasticities (OGEs) to cumulative ore production were determined on an individual mine basis. OGEs were determined by simulating mine life, iteratively changing OGE until cumulative ore production from 2017 to closure was within 5% of 2017 reserves, or until the closure year matches SNL's projected closure year if reserves were not reported. Minesite costs, transport and offsite costs, TCRC, and royalties are components of the mine's total cash cost, where the difference between the realized metal price and the total cash cost gives the total cash margin (TCM), which determines mine profitability[61]. We assume hedging profits and losses to be constant, approximating the realized metal price with cathode price. Minesite costs, transport and offsite costs, and royalties are exogenous mine parameters that were held constant throughout these simulations. Short-run ore production changes as mines alter CU, which is modeled according to Eq. (4) following a generalized method of moments dynamic panel regression model performed in previous work[47].

$$\text{Mine capacity utilization} = \begin{cases} 0.4, & \text{if in ramp up or ramp down} \\ 0.75, & \text{if not in ramp and TCM<0} \\ \text{CU}_0 * \left(\frac{\text{TCM}}{\text{TCM}_0}\right)^{0.024}, & \text{else} \end{cases} \quad (4)$$

where mine CU is assumed equal to 0.4 in ramp up and ramp down periods, which last 3 years after the opening decision and 1 year after the closing decision, respectively. While TCM less than zero indicates that the mine was not profitable in that year, mines are unlikely to shut down or halt production before the depletion of reserves. $\text{CU}_0$ was tuned such that the 2018 average CU in the simulation was equal to the 2018 average CU reported by the ICSG[62], while $\text{TCM}_0$ was assumed to be the median TCM of all operating mines in 2018[57].

Mine closure decision making is modeled based on anticipated cashflows and maximizing net present value (NPV), comparing the NPVs of entering the 1-year ramp down period in the current year or the next year, incurring reclamation costs in the following year for both cases. If the projected cash flow for the following year is calculated using the maximum value of cathode price from the preceding 5 years, NPV is calculated following Eq. (5).

$$\text{NPV} = \sum_{t=0}^{T} \frac{C_t}{(1+d)^t} \quad (5)$$

where $c_t$ is the total cashflow expected in year $t$, $d$ is the discount rate set at 10%[61,63], $T$ is 1 for ramp down beginning in year 0 and 2 for ramp down beginning in year 1, where $c_t$ is equal to the reclamation cost in year $T$. In all other years, $c_t$ is a function of cathode price, TCRC, CU, ore grade, and several mine-level exogenous variables described in previous work[47].

Mine opening decision making is significantly more complex than closure, as it is dependent on legal, political, environmental, and economic feasibility. In this simulation model, we rely solely on economic feasibility, assuming that non-economic conditions are either intrinsic to available economic data or do not act as bottlenecks to mine opening. The decision to develop a mine from the incentive pool into an operating mine is made when the internal rate of return (IRR) exceeds a cutoff value of 15%, a common guideline for assessing new mining projects[63,64]. IRR is calculated by solving for $r$ in Eq. (6).

$$\text{NPV}_r = \sum_{t=0}^{T} \frac{C_t}{(1+r)^t} = 0 \quad (6)$$

where $c_t$ includes 3 years of development capital expenditures, estimated cashflows, and reclamation costs, while $T$ is the year of mine closure. Cashflows are estimated

by simulating the lifetime of the mine using the trailing 3-year average of cathode price and TCRC. The number of mines selected from the incentive pool for opening evaluation was tuned such that annual mining production of 2018–2040 approximated a benchmark future mining supply, with cathode price and TCRC held constant. This future mine supply was calculated assuming linear growth in copper mine production, with the same growth rate as 2001–2011. This growth rate is slower than that of 2011–2018, which we assumed could not be sustained based on demand projections. The resulting time series of subsample size was held constant in all other scenarios.

**Refinery module**. We estimate cathode production from refineries as a function of cathode price, TCRC, and scrap spread, outputting concentrate demand, refined scrap demand, and cathode supply. Smelters are treated as part of the refining process. While SX-EW mines also produce copper cathode, they are included in the primary supply module and their production is directly added to total cathode production. Here primary refineries process only copper concentrate, while secondary refineries process both concentrate and scrap, with the fraction of raw material from scrap defined as the SR. We conducted generalized method of moments dynamic panel regression models of primary refinery CU, secondary refinery CU, and secondary refinery SR as functions of TCRC and No.2 scrap spread. Individual-level refinery data and TCRC from 1992 to 2016 was obtained from SNL[57] and No.2 spread from Fastmarkets AMM[65]. We model refineries as one primary and secondary refinery each for both China and RoW. Refinery production is the product of capacity and CU, where capacity was assumed to follow cathode consumption with a 1-year lag, following the direction of industry interviews. The resulting evolution of primary CU, secondary CU, and SR are described by Eqs. (7)–(9).

$$PCU_t = PCU_{t-1} \cdot \left(\frac{TCRC_t}{TCRC_{t-1}}\right)^{0.057} \quad (7)$$

$$SCU_t = SCU_{t-1} \cdot \left(\frac{TCRC_t}{TCRC_{t-1}}\right)^{0.153} \quad (8)$$

$$SR_t = SR_{t-1} \cdot \left(\frac{TCRC_t}{TCRC_{t-1}}\right)^{-0.197} \cdot \left(\frac{Spread_{No.2,t}}{Spread_{No.2,t-1}}\right)^{0.316} \quad (9)$$

where $PCU_t$ is the primary refinery CU at time $t$, $SCU_t$ is secondary refinery CU at time $t$, $SR_t$ is the secondary refinery SR at time $t$, and $Spread_{No.2,t}$ is the difference between cathode price and No.2 scrap price at time $t$. For more details, see previous work[47,56].

**Regional evolution**. Historical values for production and consumption both in China and globally were compiled from data provided by the ICSG, the International Copper Association, Minsur, Glöser et al. at Fraunhofer ISI, the International Wrought Copper Council, CRU Group, S&P Global Market Intelligence[57], Wood Mackenzie, the Shanghai Metals Market, American Metal Market, and UN Comtrade (see Supplementary Table 2 for full list of data sources and their applications).

In developing the China–RoW regional model, cathode prices, TCRC, and mining evolution were assumed to behave as global parameters due to market liquidity, while scrap spreads, copper demand, primary and secondary refined copper production and consumption, and scrap production and consumption required regionalization. China's scrap and refined copper imports were specified as exogenous variables, while concentrate imports were implicit in regional refinery operation. Scrap spreads evolve as a function of the scrap SD balance and the change in cathode price, and scrap spread elasticities to changes in these values retained their global values from the previous model. However, scrap spreads were modeled at the regional level to enable changes in regional scrap availability to impact scrap consumption and thus regional refinery evolution. The deviation of China and RoW scrap prices from the calculated global values was further explored as a model sensitivity as discussed in Supplementary Methods: Sensitivity to scrap SD elasticities. Total copper demand was previously calculated by region—China, EU, Japan, North America, and other—and was simply regrouped to reflect the China–RoW split.

These regional demand series were used to calculate China and RoW scrap generation using fabrication efficiencies for pre-consumer scrap generation and dMFA for post-consumer scrap generation. Regional, sectoral lifetimes follow lognormal distributions (lognormal($\mu$, $\sigma^2$) with $\mu = 0.1\sigma$ and are shown in Supplementary Table 6. The quantity of material reaching end of life in each year is given by Eq. (10).

$$EOL_{s,t} = \sum_{\tau=0}^{\infty} C_{s,t-\tau} F_{s,\tau} \quad (10)$$

where $C_{s,t-\tau}$ is the amount of copper entering use in sector $s$ at time $t$, $\{F_{s,\tau}\}_{\tau=0}^{\infty}$ is the lifetime distribution for products in sector $s$ in discrete frequencies, which satisfies $\sum_{\tau=0}^{\infty} F_{s,\tau} = 1$. $F_{s,\tau}$ is the fraction of final products in sector $s$ reaching end of life at year $t + \tau$, having entered use at year $t$. Therefore, $EOL_{s,t}$ is the total quantity of products from sector $s$ reaching end of life at year $t$. These regional, sectoral end-of-life quantities were converted to waste categories, then regional

collection rates and technical recycling efficiencies were applied to calculate the useable scrap generated in China and RoW. These values are given in Supplementary Table 7.

Global refinery production was previously modeled as two individual refineries representing primary and secondary refineries, where primary refineries consume only primary material and secondary refineries consume some fraction of secondary material. In secondary refineries, the SR describes the secondary fraction of total raw material consumed. In regionalizing refinery production, the two representative refineries were further split to emulate China and RoW primary and secondary refinery production. Each refinery's capacity evolved as a function of regional copper cathode demand, with long-run capacity elasticity to demand remaining the prior global value. Similarly, primary and secondary refinery CU elasticities to TCRC, as well as SR elasticities to TCRC and No.2 scrap spread, retained their global values. Initial values for regional primary and secondary CU and SR were estimated based on individual refinery data from Wood Mackenzie grouped by region, and refinery capacity was calculated such that each representative refinery's production matched 2018 regional production values based on these parameters. These methods are further elaborated in Supplementary Methods: Regional data and evolution. Refinery raw material consumption was calculated assuming concentrate to cathode and No.2 scrap to cathode efficiencies of 99%. Regional cathode consumption was calculated using the combined blending model described below.

**Copper-based product fabrication**. As in the previous, less granular model, global copper demand is estimated as the product of regional (China, EU, Japan, North America, and other) and sectoral (construction, electrical, industrial, consumer and others, and transportation) volume and intensity values. Regional sectoral volumes— e.g., total mass of all materials consumed in China's construction sector—are projected through 2040 as in prior work[47] and are static and exogenous to the model. Regional sectoral usage intensities define the copper mass per mass of material in each region and sector and are determined endogenously based on copper price and exogenous GDP per capita to allow for dematerialization, substitution, and income effects. Bayesian regression models were used to estimate the parameters in this model, shown in Eq. (11), as described by Bhuwalka et al.[45].

$$\Delta \log (I_{s_i, r_j, t}) = \beta_{0, s_i} + \beta_{s_i} \Delta \log (P\prime_t) + \beta_{r_j} \Delta \log (P\prime_t) + \beta_{GDP} \Delta \log (GDP_{r_{j,t}}) \quad (11)$$

where each sector, $s_i$, and region, $r_j$, has copper use intensity at time $t$ represented by $I_{s_i, r_j, t}$, which is a function of a sector-specific intercept $\beta_{0, s_i}$ representing dematerialization, sector- and region-specific copper price coefficients $\beta_{s_i}$ and $\beta_{r_i}$ where $P\prime_t = \frac{P_{t-1} + P_{t-2}}{2}$ is the first lag of trailing 2-year average cathode price, and $\beta_{GDP}$ representing the intensity response to regional GDP per capita. For more details, see our previous work[45,47,56].

The resulting regional sectoral demand values were then converted to regional demand by shape (e.g., copper or alloyed wire or tube) using global parameters based on data from Glöser et al., which permitted distinction between alloyed and unalloyed products, including refined copper use at the global scale. Unalloyed products were assumed on average >99.8% Cu by mass, while alloyed product compositions were determined based on Copper Development Association supplier databases and industry expert interviews as described in Supplementary Methods: Semi-fabricator alloy distribution framework. As such, alloyed semi-fabricator production was broken into 190 representative Unified Numbering System alloys, noting European Committee for Standardization equivalents. It was assumed that the overall distribution of alloying elements within each shape remains constant over time, and consequently the fraction of each shape occupied by each alloy was held constant for each year of production. Since each sector is composed of different fractions of each shape, and that sectors evolve independently as shown in Eq. 1, demand for individual alloys does not, however, remain constant. Alloy compositional requirements across eight impurity elements were considered for the blending component of the linear programming optimization model described in the following section.

Scrap generation was broken down into 14 categories representing the most common ISRI post-consumer scrap grades and 191 categories representing post-industrial scrap produced by alloyed semi-fabricators, with 190 categories representing alloyed post-industrial scrap and the single additional alloy representing unalloyed post-industrial scrap. Annual post-consumer scrap generation values for China and RoW were calculated using standard dMFA methods with lognormal sectoral lifetime distributions, with sectoral collection rates and recycling efficiencies for both regions. Refined metal markets were assumed to be sufficiently liquid to permit any quantity to be consumed at the same unit price globally, but given availability concerns associated with post-consumer scrap consumption, post-consumer scrap prices were determined using an order book formulation, where average purchasing prices increased with total quantities consumed within each region. Given that inventories were sufficiently large that each scrap grade was not fully consumed each year, this formulation permitted an increase in regional scrap availability in a given year to produce an increase in scrap consumption in that region. The resulting equations describing scrap prices relative to quantities consumed were formulated for each scrap grade as functions of scrap availability and scrap price, which was determined using econometric time series analysis. Following the assumption that scrap markets are

illiquid, scrap prices were allowed to change based on regional SD imbalances. Additional details surrounding order book, scrap price formation, and additional data for scrap generation are described in Supplementary Methods: Scrap price, availability, and their interplay.

With regional manufacturer production and scrap availability determined at the compositional level, a linear programming optimization model was developed within Gurobi Optimization software[66], where composition and production quantity constraints were imposed and price was minimized. This model was further constrained to consume the total quantity of refined copper determined at the global level above, with the distribution between China and RoW a result of scrap and refined metal prices and determined by the optimization model (see Supplementary Methods: Linear programming optimization model).

**Life cycle assessment**. This work was primarily performed using the Ecoinvent 3 database within the software package SimaPro, using the environmental damage indicators offered by the Tool for the Reduction and Assessment of Chemical and other Environmental Impacts (TRACI) 2.1 midpoint life cycle inventory analysis method, which provides the following ten damage categories: Ozone depletion (kg CFC-kk), Global warming (kg $CO_2$ eq), Smog (kg $O_3$ eq), Acidification (kg $SO_2$ eq), Eutrophication (kg N eq), Carcinogenics (CTUh), Non carcinogenics (CTUh), Respiratory effects (kg PM2.5 eq), Ecotoxicity (CTUe), Fossil fuel depletion (MJ surplus). We also considered total energy consumption using Cumulative Energy Demand V1.11 and water use following Berger et al. For mines, we calculated $CO_2$e emissions, water use, and energy consumption as a function of ore grade according to relationships established by Northey et al. Average regional ore grades for concentrate and SX-EW mines were calculated using SNL data, average regional $CO_2$e emissions, water use, and energy consumption were calculated following Eq. (12), and the resulting values were multiplied by regional scaling factors (Supplementary Table 11) to reach the regional values from Ecoinvent 3, TRACI 2.1, Cumulative Energy Demand V1.11, and Berger et al. (Supplementary Table 12).

$$C = R_r A g^B \qquad (12)$$

where $C$ is the calculated impact for the category applied, $A$ and $B$ are empirically determined constants using a power series trendline on global concentrate and SX-EW mine data for $CO_2$ emissions, water consumption, and energy use from Northey et al. with values in Supplementary Table 13[67], and $g$ is the ore grade of the mine[57]. Additional regional scaling factors, $R_r$, were implemented such that the calculated average ore grade for that region (from SNL, Supplementary Table 11) produced the regional unit $CO_2$ emissions, energy, or water value for Ecoinvent processes for copper concentrate (sulfide ore) or copper (from SX-EW). Regions include Oceania, Africa, Europe, North America, China, other Asia, and Latin America. Further details are described in previous work[47]. These processes were used to calculate the emissions or consumption for each mine individually, where the remaining nine TRACI indicators were calculated based on the ratio of $CO_2$e emissions to each indicator for the respective region in Ecoinvent 3. We assume that the global distribution of mining activities remains constant from 2018 through 2040.

Regional SX-EW mining environmental impacts were determined using the SimaPro global value for copper from SX-EW, scaled by the impacts of regional concentrate mining relative to global average concentrate impacts. Due to the exponential nature of the mine impact calculations, individual mine unit impacts were capped according to the highest values from Northey et al. for concentrate mines[67], with caps of 10 kg $CO_2$/kg Cu, 150 MJ/kg, and 0.27 m$^3$ water/kg. With less data available for SX-EW mines, we scaled these values by the ratio between the highest calculated regional SX-EW mine impact and the highest regional concentrate mine impact for each impact category, giving maxima of 28 kg $CO_2$/kg Cu, 450 MJ/kg, and 0.27 m$^3$ water/kg.

Impacts for the remaining supply chain components follow values from Ecoinvent 3, TRACI 2.1, Cumulative Energy Demand V1.11, Berger et al.[68], Giurco et al.[69], and Chen et al.[70]. Here primary refining includes smelting impacts as well, and both primary and secondary refining are treated at the regional level. Direct melting of scrap is treated globally due to data limitations but is a function of scrap grade (Supplementary Table 15). Semi-finished goods manufacturing unit impacts are semi-regional, with global impacts from wire drawing on ∼56% of production based on data from Glöser et al.[46]. The remainder is treated using metal working impacts, with Europe and North America using one set of impacts and other regions using impacts for RoW (Supplementary Table 17). For all of these impacts, we assume the regional distribution within RoW remains constant, with the fraction of primary refining, secondary refining, direct melt scrap consumption, and manufacturing occurring in China and RoW being results of model evolution. We assume that refining impacts do not change as functions of ore grade and that manufacturing and scrap direct melting impacts do not change over time, in alignment with previous studies[71].

We modeled regional changes in $CO_2$e emission intensity of electricity generation following the U.S. Energy Information Administration's reference case, which projects a 14% decline globally from 2018 to 2040 (Supplementary Table 20)[35]. This reference case indicates a 22% decline in global $CO_2$e emission intensity of electricity generation using the 2010–2050 range of Ciacci et al.[71], which falls between their "market rules" (10% decline) and "toward resilience" (32% decline) scenarios. Industrial sources generally state that comminution

accounts for at least 50% of mine energy consumption, while academic studies have reported 46% of copper mine $CO_2$ emissions from comminution[72], 48–62% of mine energy as electricity[67], and 70% of all mine energy for comminution[73]. With comminution requiring electricity for the operation of mechanical grinding apparatuses, we assume that 55% of copper mining activity energy consumption is in the form of electricity, with the remainder being primary energy unaffected by decreased emission intensity of electricity. Annual $CO_2$e emissions calculated using the original parameters were scaled according to the regional emission intensity change relative to 2018.

## Data availability
The data that support the findings of this study are available from the corresponding author upon request. Source data are provided in the figshare repository https://doi.org/10.6084/m9.figshare.14390489.v3[74].

## Code availability
The code supporting the findings of this study is available in the figshare repository https://doi.org/10.6084/m9.figshare.14390489.v3[74], with additional detail available upon request.

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

## Acknowledgements

The views expressed in this article are those of the authors alone. The authors acknowledge funding from the National Science Foundation Award #1605050, CBET program that provided support to make this work possible. This research was conducted on the traditional, unceded territory of the Wampanoag Nation. We acknowledge the painful history of forced removal from this territory, and we respect the many diverse indigenous people connected to this land.

## Author contributions

X.F. and E.A.O. formulated the study of China's solid waste import ban and the underlying copper material system model. J.R. formulated scenarios for the solid waste import ban, refined import changes, and supply chain shocks and performed the requisite analysis and graphical representation with substantive input and feedback from X.F. and E.A.O. J.R. also developed the regional variant of this model, the semi-fabricator alloy distribution framework, the scrap price-availability model, and the linear

programming optimization model based on the work of X.F. The copper demand model was formulated and developed by K.B. with R.R., and X.F. implemented it within the model. J.R. developed the environmental impact analysis under the guidance of E.A.O. The writing of the manuscript was led by J.R. and E.A.O. with substantive input from all other authors, and figures were created by J.R. with feedback from E.A.O. and X.F. Icons in Fig. 1 were created by prettycons, Freepik, monkik, Smashicons, Good Ware, and srip from flaticon.com and adapted by J.R.

## Competing interests

The authors declare no competing interests.
