## [Peer Review File · Nature Communications]

REVIEWER COMMENTS

Reviewer #1 (Remarks to the Author):

Review of "Emission impacts of supply chain disruptions for COVID and China's solid waste import ban"

Recommendation: Major revisions and re-review

Overall comments

The manuscript presents interesting projections until 2040 regarding China's copper supply chain disruptions as COVID and its relations to global economic crisis and environmental impacts. Furthermore, the Supplementary Information proves the extensive work. Nevertheless, the paper currently addresses the discussion insufficiency in sections to exploit its novel aspect adequately and requires considerable revisions to improve the quality of work. This is motivated and explained below: The research focuses on copper, thus, copper supply chain should be included in the title. The present version of the title does not excavate the whole output of work. I suggest to revise the title according to all inclusions.

In the study, assessments and projections of copper supply chain are based on a considerable number of data sources. Authors have also concluded that they confirm results of some published articles regarding China's domestic copper scrap generation: Zeng,¹⁹ Dong,²² Eheliyagoda,⁴¹ Liu,⁴² and Wang²⁰. This is interesting but may raise questions whether this study is developed to support existing publication output. Therefore, it is essential to further clarify the originality and novelty of the study. The present form does not have sufficient arguments to justify the novelty of the paper. Authors should explicitly specify the novelty of their work. What progress against aforementioned studies was made in this study? Mention this in the revised manuscript.

At the current form, the research method is confused with the results section. This should be re-corrected appropriately presenting method descriptions solely in the methods section. It is recommended to explain the methodology with a flow chart design framework and description. The manuscript does not contain a separate discussion section. I suggest to restructure the article with a discussion section. Under the discussion section, it is also recommended to discuss and explain what the appropriate policies should be based on the findings of this study. It is strongly recommended to add practical implications of this study, outlining the challenges in the current research, future work, and recommendations under the discussion section.

Please write the full term of "RoW" at the first paragraph of each section if authors use it for explanations there, otherwise it is difficult to link this word with the explanation in each section. It is recommended to provide a nomenclature or abbreviation table/box at the start of the Supplementary Information, explaining all the abbreviated words used throughout the manuscript and Supplementary Information.

It is suggested to add a table/appropriate entity in the Supplementary Information clearly stating each ban definition used in the study.

It is recommended to state used assumptions for scenario analyses point by point in the Supplementary Information.

Xianlai Zeng

Reviewer #2 (Remarks to the Author):

I'd advise against the publication of this manuscript. On the one hand, I'm sympathetic with the goals the authors aim to achieve – to model how (a) changes in China's important policies and (b) the COVID-19 pandemic affect the copper sector and its environmental impacts.

That said, I do not find this paper to be effective or particularly clear in terms of achieving those goals. On the plus side, the paper is well-written. But the main body of the paper presents the results of simulations with essentially no information of the structure of the model and its

conceptual/empirical foundations. The paper's methods section comes after the conclusions. While that matches Nature's style, there's just not a way to understand the approach without perusing the details of the post-conclusions sections of the manuscript and the supplementary material.

From there, I'm just not impressed by the material on pp. 17-26. The discussion is discursive and simply not clear in understanding each step of the analysis in terms of equations and approach to estimating to model. It's the kind of presentation one would expect from a grey literature report aimed at non-specialists. I also believe that pp. 17-26 rely too heavily on citations to Xinkai Fu's apparently unpublished doctoral dissertation, which is identified on line 329 as "the basis of this model." That fine and good, but the authors' aim here is basically to publish a policy-oriented paper in a super-visible, generalist journal. That, I would say, needs to be backed by publications in more specialized journals – most likely the Journal of Cleaner Production – that can validate the modeling framework. In my view, Nature Communications is not the right outlet for this kind of deep spade work.

I'd also say that it seems a bit premature to try to measure the long-run impacts of changes in China's trade policies, which at this stage has a history of just 2-3 years. Adding the COVID pandemic to the picture is just another complication. That's an attempt to achieve too much in just one paper using a plurality of methods that aren't that well explained.

Finally, I'm not that impressed by the paper's sources and citations. There's an overemphasis on grey literature and sources that are focused mainly on news or current events. The technical citations go especially to applied journals in areas like industrial ecology and recycling. Though the paper aims to apply methods from econometrics, the paper's engagement with sources from economics and econometrics is slight even though the research questions explored by the authors have strong economic dimensions.

Reviewer #3 (Remarks to the Author):

The manuscript "Emission impacts of supply chain disruptions for COVID and China's solid waste import ban" is a nice piece of work that addresses possible supply-chain reactions stemming from the China solid waste import ban, the related environmental impacts, and mechanisms for maximizing environmental benefits.

The manuscript is well written and figures are informative. The model created is relatively complex but seems to be robust and reliable. The authors provide considerable underlying information and dataset are clearly reported.

My only concern with this work relates to the modelling of carbon intensity of electricity generation. It is not clear to me if (and how) the authors modelled the possible evolution of national/regional/global electricity production mix to 2040. Considering energy inputs in copper refining, the effect of decarbonization of electrical energy may play an important role in increasing environmental benefits, particularly in the copper cycle, as shown in a relevant recent paper (Ciacci et al., 2020).

If the authors have included this parameter in their model, I would suggest to add a description of the main assumptions in the text. If not, I think that they should reconsider their model and include exogenous variables to capture that effect. As the authors may know, the International Energy Agency has developed accurate scenarios for electricity generation mix at different geographical scale which may be helpful for this work.

Likewise, Have the authors distinguished between smelting technology and efficiency by country/region? For instance, more than 70% of total copper processed in Europe is refined in plants equipped with best available techniques (BATs). What is the situation in the rest of the world? How

might a progressive BATs implementation reduce the environmental impacts associated with copper smelting and refining?

I think that this manuscript would benefit from a more extensive discussion on the issues raised above.

Cited article:

L. Ciacci, T. Fishman, A. Elshkaki, T.E. Graedel, I. Vassura, F. Passarini, Exploring future copper demand, recycling and associated greenhouse gas emissions in the EU-28, *Global Environmental Change*, 63, 2020, 102093, doi.org/10.1016/j.gloenvcha.2020.102093.

Response to Referees

Dear Dr. Kyle Frischkorn and Referees,

Thank you for considering our manuscript in *Nature Communications* and for your detailed comments, feedback, and suggestions. They have been very helpful in revising our manuscript. We provide a numbered list of reviewer feedback separated based on general themes, and the resulting table is below. If there are further changes you would like us to make, we would be happy to implement them as well. Reviewer comments appear in the column on the left, and our responses appear in the column on the right. In the revised manuscript, changes and additions appear using the Track Changes functionality in Microsoft Word, and are underlined and given a different color.

Reviewer 1

1) Copper supply chain should be included in the title	We have updated the title to reflect this feedback.
2) In the study, assessments and projections of copper supply chain are based on a considerable number of data sources. Authors have also concluded that they confirm results of some published articles regarding china's domestic copper scrap generation: Zeng,19 Dong,22 Eheliyagoda,41 Liu,42 and Wang20. This is interesting but may raise questions whether this study is developed to support existing publication output. Therefore, it is essential to further clarify the originality and novelty of the study. The present form does not have sufficient arguments to justify the novelty of the paper. Authors should explicitly specify the novelty of their work. What progress against aforementioned studies was made in this study? Mention this in the revised manuscript.	We have added detail in discussion and introduction sections addressing this issue and appreciate these comments for their contribution.
3) At the current form, the research method is confused with the results section. This should be re-corrected appropriately presenting method descriptions solely in the methods section. It is recommended to explain the methodology with a flow chart design framework and description.	We have made significant updates in the methods section that we hope will make it more clear, and a detailed flow chart has been placed in the supplementary information.
4) The manuscript does not contain a separate discussion section. I suggest to restructure the article with a discussion section. Under the discussion section, it is also recommended to discuss and explain what the appropriate policies should be based on the findings of this study. It is strongly recommended to add practical implications of this study,	We have changed the header of the "Conclusions" section to "Discussion," which aligns better with Nature Communications' style requirements as well. In this section, we have incorporated suggestions for future work, opportunities for improvement, and perceived challenges in future research. We believe this suggestion

outlining the challenges in the current research, future work, and recommendations under the discussion section.

5) Please write the full term of “RoW” at the first paragraph of each section if authors use it for explanations there, otherwise it is difficult to link this word with the explanation in each section. It is recommended to provide a nomenclature or abbreviation table/box at the start of the supplementary information, explaining all the abbreviated words used throughout the manuscript and supplementary information.

6) It is suggested to add a table/appropriate entity in the supplementary information clearly stating each ban definition used in the study.

7) It is recommended to state used assumptions for scenario analyses point by point in the supplementary information.

has significantly improved this section and appreciate this suggestion.

We have added an abbreviation table in the supplementary information and written out “rest of world” in its first use of each section.

We have included the resulting table in the methods section, as we found this suggestion helpful in clarifying scenario definitions.

We have added a bulleted list of assumptions in the supplementary information.

Reviewer 2

1) The main body of the paper presents the results of simulations with essentially no information of the structure of the model and its conceptual/empirical foundations. The paper’s methods section comes after the conclusions. While that matches nature’s style, there’s just not a way to understand the approach without perusing the details of the post-conclusions sections of the manuscript and the supplementary material.

2) The material on pp. 17-26. The discussion is discursive and simply not clear in understanding each step of the analysis in terms of equations and approach to estimating to model.

3) I also believe that pp. 17-26 rely too heavily on citations to Xinkai Fu’s apparently unpublished doctoral dissertation, which is identified

We are unsure whether we can address this feedback without violating Nature’s requirements, but we would be happy to make further modifications to the structure as editorial policy allows.

We have added more thorough description in the methods section, including a description of scenarios, price formation, mine evolution, and refinery behavior. We have also added a supplementary figure that provides more detail on the model evolution pathway and input data associated with each step. We believe this feedback has significantly improved the methods section and we appreciate this contribution.

We have added detail around mining, refining, and price evolution to the revised manuscript. We intend to publish the underlying model

on line 329 as “the basis of this model.” That fine and good, but the authors’ aim here is basically to publish a policy-oriented paper in a super-visible, generalist journal. That, I would say, needs to be backed by publications in more specialized journals – most likely the journal of cleaner production – that can validate the modeling framework. In my view, nature communications is not the right outlet for this kind of deep spade work.

4) I’d also say that it seems a bit premature to try to measure the long-run impacts of changes in china’s trade policies, which at this stage has a history of just 2-3 years. Adding the COVID pandemic to the picture is just another complication. That’s an attempt to achieve too much in just one paper using a plurality of methods that aren’t that well explained.

5) Finally, I’m not that impressed by the paper’s sources and citations. There’s an overemphasis on grey literature and sources that are focused mainly on news or current events. The technical citations go especially to applied journals in areas like industrial ecology and recycling. Though the paper aims to apply methods from econometrics, the paper’s engagement with sources from economics and econometrics is slight even though the research questions explored by the authors have strong economic dimensions.

framework in the Journal of Industrial Ecology, and the corresponding manuscript is included in our resubmission documents.

While it is true that both COVID and China’s import ban are ongoing events, one can (and should) still try to understand and quantify their implications. The implications will become clearer as the market evolves and as we have more information, and this issue should be a motivation for further studies when this information becomes available. We hope to demonstrate the breadth of capabilities these methodologies offer, with the goal of permitting more granular analysis of these issues in the future and providing a basis for future work.

Additionally, we have reorganized and added to the methods in a way that we hope addresses the limited detail in the prior submission and also provides methods that will be useful in those future studies.

With the additional methods included in this paper, we have added citations from econometrics. Additionally, our reliance on grey literature is a result of academic literature lacking timely and granular information. With the granularity and bottom-up nature of this model and the emphasis on investigating current events, we need to rely on sources outside academia. In doing so, we aim to fill a gap in the academic literature.

Reviewer 3

1) My only concern with this work relates to the modelling of carbon intensity of electricity generation. It is not clear to me if (and how) the authors modelled the possible evolution of national/regional/global electricity production mix to 2040. Considering energy inputs in copper refining, the effect of decarbonization of electrical energy may play an important role in increasing environmental benefits, particularly in the copper cycle, as shown in a relevant recent paper (Ciacci et al., 2020, doi.org/10.1016/j.gloenvcha.2020.102093). If the authors have included this parameter in their model, I would suggest to add a description of the main assumptions in the text. If not, I think that they should reconsider their model and include exogenous variables to capture that effect. As the authors may know, the international energy agency has developed accurate scenarios for electricity generation mix at different geographical scale which may be helpful for this work.

2) Likewise, have the authors distinguished between smelting technology and efficiency by country/region? For instance, more than 70% of total copper processed in Europe is refined in plants equipped with best available techniques (BATs). What is the situation in the rest of the world? How might a progressive bats implementation reduce the environmental impacts associated with copper smelting and refining?

We have adjusted our results according to the EIA's reference scenario for regional carbon dioxide emissions intensity of electricity generation. This was a particularly valuable suggestion given the sensitivity of results to relative regional emissions intensities, where rapid decarbonization within China serves to decrease the severity of emissions effects associated with the solid waste import ban.

We have distinguished between environmental impacts of smelting, refining, and mining technologies by region in this work, with more detail added in the methods to describe this granularity. Our other study nearing publication, attached within the resubmission documents, covers the basis of this model and addresses these bigger-picture questions due to its emphasis on material efficiency strategies as opposed to regional policy changes. We address BATs in Figure 5 of our other manuscript, attached in the resubmission documents.

REVIEWERS' COMMENTS

Reviewer #1 (Remarks to the Author):

The manuscript is improved with some screwing logic and data. The following weak points could be considered:

1. The authors do not know the detailed information on China's ban policy on solid waste import. Actually, China started the ban policy of solid waste import in the year 2000. At the initial time, the main type with waste import ban was hazardous waste. In July 2017, some secondary resource types were also involved in this policy. In this manuscript, there are many places, like Line 94-95, that should be improved again.

please read the published article Environ. Sci. Technol. 2018, 52, 14, 7595–7597

<https://doi.org/10.1021/acs.est.8b01852>

2. Wrong use for life cycle analysis (LCA) with life cycle assessment. Based on the results, I believe it could be a life cycle assessment.

3. I did not find the effective utilization of dynamic material flow analysis (dMFA). Please make formal procedures.

Reviewer #3 (Remarks to the Author):

The authors have addressed all my major concerns, I have no further comments. However, I would suggest to double check abbreviations, they should be defined at the first use in the text: Rest of the world (RoW) is defined at least three times in the manuscript.

Reviewer #1

The manuscript is improved with some screwing logic and data. The following weak points could be considered:

1. The authors do not know the detailed information on China's ban policy on solid waste import. Actually, China started the ban policy of solid waste import in the year 2000. At the initial time, the main type with waste import ban was hazardous waste. In July 2017, some secondary resource types were also involved in this policy. In this manuscript, there are many places, like Line 94-95, that should be improved again. please read the published article Environ. Sci. Technol. 2018, 52, 14, 7595–7597 <https://doi.org/10.1021/acs.est.8b01852>

2. Wrong use for life cycle analysis (LCA) with life cycle assessment. Based on the results, I believe it could be a life cycle assessment.

3. I did not find the effective utilization of dynamic material flow analysis (dMFA). Please make formal procedures.

Thank you for catching this significant oversight, we have updated the referenced section and Figure 2a to reflect corrected policies.

You are correct, that was an oversight and we appreciate you catching this mistake.

This comment helped to show that overall, the treatment of scrap generation methods was lacking, and we have added to the regional evolution section in the methods to describe the dMFA application and regional scrap generation calculations. Given the emphasis on scrap availability in this work, we believe this point is a substantial contribution.

Reviewer #3

The authors have addressed all my major concerns, I have no further comments. However, I would suggest to double check abbreviations, they should be defined at the first use in the text: Rest of the world (RoW) is defined at least three times in the manuscript.

Thank you, we have double checked abbreviations and removed several definitions of RoW such that it is defined only at the first use in the text and when required in a figure caption.